# Chirality-controlled crystallization via screw dislocations

Baeckkyoung Sung[1], Alexis de la Cotte[1] & Eric Grelet [1]

Chirality plays an important role in science from enantiomeric separation in chemistry to chiral plasmonics in nanotechnology. However, the understanding of chirality amplification from chiral building blocks to ordered helical superstructures remains a challenge. Here, we demonstrate that topological defects, such as screw dislocations, can drive the chirality transfer from particle to supramolecular structure level during the crystallization process. By using a model system of chiral particles, which enables direct imaging of single particle incorporation into growing crystals, we show that the crystallization kinetic pathway is the key parameter for monitoring, via the defects, the chirality amplification of the crystalline structures from racemic to predominantly homohelical. We provide an explanation based on the interplay between geometrical frustration, racemization induced by thermal fluctuations, and particle chirality. Our results demonstrate that screw dislocations not only promote the growth, but also control the chiral morphology and therefore the functionality of crystalline states.

---

[1] Centre de Recherche Paul-Pascal, CNRS & Université de Bordeaux, 115 Avenue Schweitzer, 33600 Pessac, France. Correspondence and requests for materials should be addressed to E.G. (email: grelet@crpp-bordeaux.cnrs.fr)

Chirality, defined as the lack of mirror symmetry, was first evidenced by Pasteur, when he isolated left-handed from right-handed sodium ammonium tartrate crystals[1]. For chiral molecules, crystallization is the most widely used sorting strategy for separating their two enantiomers[2]. Harnessing crystal handedness has a major role in materials science and nano-technology, from asymmetric catalysis to optical sensing, chiral plasmonics, and biomineralization[3–5]. Despite its importance, the selection of crystal chirality during its growth is not yet fully understood and controlled[6–8]. More generally, the understanding of chirality transfer from chiral primary units, molecules or particles, to ordered helical superstructures, as usually found in liquid crystals, secondary and tertiary structure of proteins and nucleic acids, supramolecular polymers and fibers, nanowires and nanocrystals, remains largely puzzling[9,10]. Therefore chiral selection methods to tune the handedness of self-assembled superstructures and of crystals from racemic to homohelical are a topical challenge[3,10–14].

Here, we show that introducing screw dislocation defects promotes the formation of helical crystals, whose handedness is tunable and driven both by growth rate and chirality of the constituent particles. By using a model system of chiral colloidal rod-shaped particles, in situ visualization of the crystallization and of the defects is performed, revealing either helical or achiral one rod-length-thick hexagonal platelets. We demonstrate an original mechanism of chirality transfer mediated by screw dis-locations and monitored by the kinetics of growth. If screw dis-locations have been shown to favor the crystal growth, following the pioneering work of Franck[15], as well as more recently to drive the one-dimensional (1D) or two-dimensional (2D) growth of nanowires and nanostructures[13,16–19], we evidence here an additional role of these ubiquitous topological defects: monitoring and tuning the chiral morphology and therefore the functionality of crystalline materials[3,20].

## Results

**Hexagonal crystalline platelets.** As many physical processes such as crystallization do not fundamentally depend on the size of the constituent units, colloidal particles thought as giant atoms interacting via entropic excluded volume and electrostatic repulsion, have been widely used to better understand many molecular systems[21,22]. Following this idea we chose 1 μm long filamentous viruses, called fd, as model system of elongated particles that can be visualized and tracked by optical micro-scopy[23,24]. Specifically, these rod-like viruses have been used as building blocks in model liquid crystal systems, as well as to create functional materials[25]. In order to tune particle interaction from repulsive to attractive, non-adsorbing polymers are added to the dilute isotropic liquid fd suspensions and act as condensing agent of the viruses, which aggregate side-by-side into different morphologies, such as the recently reported liquid crystalline colloidal membranes[26,27]. In our work, the depletion attraction is induced by a relatively small molecular mass polymer (see Methods) resulting in the formation of 2D crystalline lamel-lae[23,28,29] (Fig. 1a). These platelets are composed of a single layer of hexagonally packed aligned viral rods and of a long central crossing spindle-like defect (Fig. 1b). Each layer is either partially overlapped (Fig. 1b, c) or completely flat (Fig. 1a, d, e). It exhibits in both cases an internal crystalline ordering as shown by long-range positional order probed by small-angle X-ray scattering (SAXS, Supplementary Figure 1) and also confirmed by the absence of any detectable particle self-diffusion within the structure (Supplementary Movies 1 and 2). The depletion attraction is maximized when the excluded volume between the colloids is the largest, implying both that longer rods are more likely to coalesce than shorter ones, and that lateral association of rod-shaped particles is the most favored. For these reasons, we observed first the formation of thin columnar bundles which are a few micrometers long (Fig. 1b) and have a diameter of about 0.3 μm, composed of both multimeric viruses (Fig. 1e, f), having a higher contour length and present in small fractions in the sus-pensions (Supplementary Figure 2), and of single fd (Fig. 1f). In a second step, these protruding filaments act as nucleation sites for

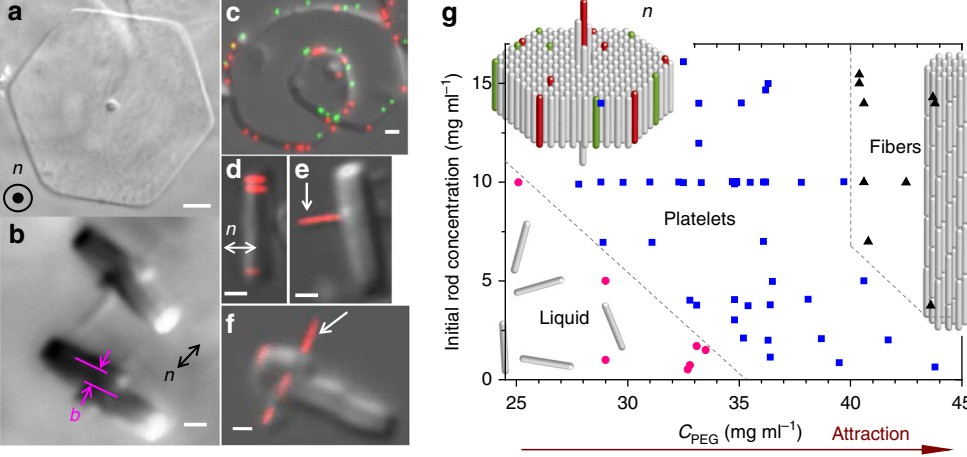

**Fig. 1** Self-assembly of rod-like viruses by effective depletion attraction into two-dimensional crystalline colloidal platelets. **a** Face-on view by differential interference contrast (DIC) optical microscopy of a hexagonal platelet. The platelet main axis is indicated by *n*. **b** Side-on DIC image of two platelets penetrated by a long central columnar filament around which a screw dislocation with a Burgers vector of magnitude *b* ≈ 1 μm has grown, resulting in some partial overlap of the one-rod-length-thick monolayer. **c** Helical platelet where the axial dislocation is seen from top-view, and decorated by red and green fluorescently labeled viruses at its edges. Observation performed by overlaying DIC and fluorescence images. **d** Composite DIC/fluorescence microscopy side-on image showing that platelets are composed of aligned fd rods within the whole crystalline structure, lying parallel to the platelet main axis, *n*. The addition of a low fraction of fluorescently labeled viruses evidences the absence of tilt and therefore of twist at the platelet edges. **e-f** Fluorescence microscopy combined with DIC reveals that the central protruding needle-like core is composed of both single viruses and the low fraction of multimeric viruses (red dimers, as indicated by the white arrows) initially present in the suspension. Scale bars, 1 μm. **g** The range of stability of hexagonal platelets is located between the isotropic phase of rods (fd-wt strain) and columnar fibers, by increasing depletion attraction

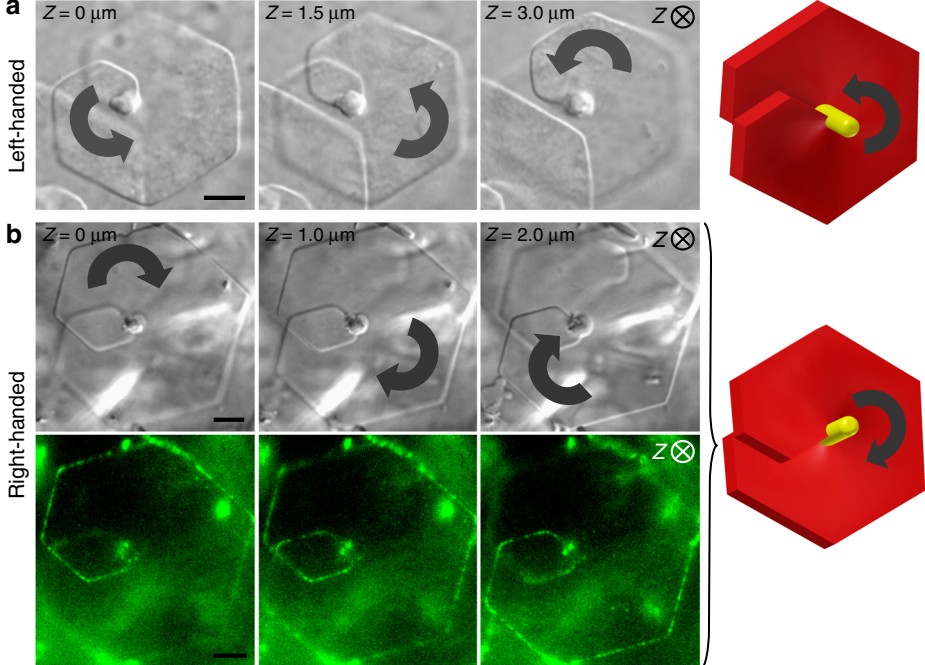

**Fig. 2** Hexagonal platelets with opposite screw dislocation handedness as shown by optical microscopy by capturing images at different focal depths $Z$ (along the platelet main axis, $n$) and corresponding schematic representations. **a** Left-handed screw dislocation (DIC, Supplementary Movie 3). The handedness is indicated by a black curved arrow. **b** Right-handed screw dislocation (upper: DIC, Supplementary Movie 4; lower: fluorescence, Supplementary Movie 5) for which the hexagonal platelet edge is decorated with fluorescently labeled rods. Scale bars, 2 μm

the growth of one rod-length thick hexagonal crystalline monolayers of aligned rods. All platelets were found to have this long defect near their center[30], resulting either in flat and achiral platelets, or in helical ones for which single screw dislocations are clearly evidenced both from side-on (Fig. 1b, f) and face-on (Figs. 1c and 2) views. The screw dislocations use these protruding bundles as central cores, and they are found to have either left-handedness (Fig. 2a, Supplementary Movie 3) or right-handedness (Fig. 2b, Supplementary Movies 4 and 5), as directly determined by varying the optical microscope focus through the sample thickness. The Burgers vector characterizing the dislocation is along the platelet main axis $n$ and has a magnitude of $b \approx 1$ μm (Fig. 1b), corresponding to the monolayer thickness and also to the lattice constant in this direction.

The stability range of such hexagonal platelets is provided in Fig. 1g, and these 2D crystals are the first self-assembled structures appearing by increasing depletion attraction from the isotropic liquid phase of rods. No other morphology as liquid-like colloidal membrane exists at lower rod attraction. By further increasing the polymer concentration, 1D growth is promoted with respect to the 2D one, and columnar fibers are obtained in which rods also show hexagonal long-range positional order, as probed by SAXS (Supplementary Figure 1).

**Kinetics of growth and topological defects**. The kinetics of the platelet growth is quantitatively determined in Fig. 3a and is shown in Supplementary Movies 6 and 7. The radial growth of the monolayer is preceded by the formation of the central core defect, for which the induction time to nucleate varies from a few seconds to weeks according to the initial conditions of the sample. The 2D crystallization of the platelets has been visualized at the single particle level, with the in situ imaging of an attachment event of a rod-like particle from the isotropic liquid to the platelet edge (Fig. 3b, Supplementary Movie 8). Quantitatively, the 2D growth is characterized by the growth rate $k_r$, after fitting

experimental data with a logistic function (See Eq. (1) detailed in Methods). This accounts both for an exponential increase of the platelet size in the early stage and for a saturation at long time stemming from the finite number of rod-like particles in the sample (Fig. 3a). Up to three orders of magnitude of growth rates $k_r$ have been probed, corresponding to a time scale from a few minutes for the fastest crystallizations to weeks for the slowest ones. The crystallization is faster at fixed depletion attraction by increasing the initial rod concentration, as expected from classical crystal growth theory when the degree of supersaturation increases[31]. Two distinct platelet morphologies have been distinguished as a function of the growth rate: a majority of flat achiral 2D crystallites at low $k_r$, and helical platelets with the presence of a single screw-dislocation for faster growth, as quantitatively reported in Fig. 3c. As initially pointed out by Frank[15], screw dislocation defects catalyze crystal growth by creating a step-edge promoting the interaction at the liquid-solid interface. This non-vanishing step can induce a self-perpetuating spiral growth, which fastens the crystal growth. Beyond spiral stepped crystal facets[32], screw dislocation driven growth has been shown to drive the formation of 1D nanomaterials such as nanowires[16–18], as well as more recently of 2D inorganic crystals[19,28]. Figure 3c indicates that the ratio of platelets exhibiting a screw dislocation over the total number of crystallites increases with the growth rate, and it confirms that dislocation assisted growth can be extended to colloidal systems beyond the seminal prediction by Burton–Cabrera–Frank for atomic and molecular crystals[31].

**Screw dislocation handedness and enantiomeric excess**. As the handedness of each screw dislocation can be determined in situ by optical microscopy (Fig. 2 and Supplementary Movies 3–5) for big enough platelets (see Methods), the enantiomeric excess ee = $(R-L)/(R+L)$ where $R$ and $L$, respectively, stand for the number of platelets having a right-handed or a left-handed screw

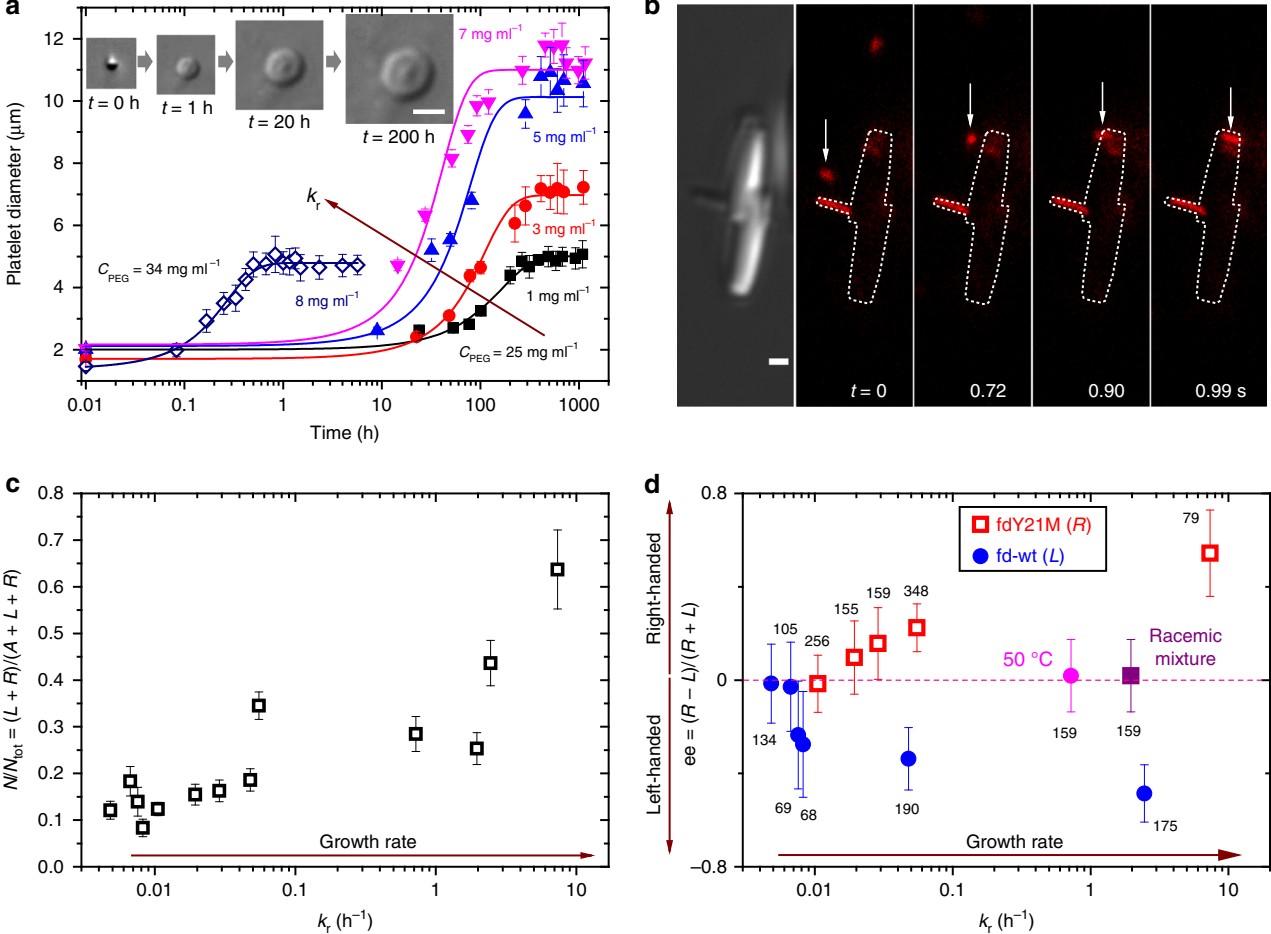

**Fig. 3** Platelet screw dislocation handedness monitored by the kinetics of growth. **a** Radial growth of the hexagonal platelets characterized by the growth rate $k_r$ defined according to Eq. (1). $k_r$ increases both with the initial viral rod (fdY21M strain) concentration and with the depleting polymer concentration (varied from 25 to 34 mg ml$^{-1}$). The kinetics of growth is obtained from DIC image sequences as shown in insets (Supplementary Movies 6 and 7). Error bars: standard deviation. Scale bar, 2 μm. **b** Observation at the single particle level of the 2D growth of a helical platelet seen by DIC microscopy. A fluorescently labeled rod (tracked by a white arrow) is shown to diffuse in the liquid phase before attaching at the platelet edge (Supplementary Movie 8). For clarity, the outline of the platelet is indicated in the fluorescence images (white dotted line). **c** The ratio of chiral and antichiral platelets, $N = L + R$, exhibiting screw dislocation, over the total number of platelets $N_{tot}$ (including the achiral ones, $A$), increases with the kinetics of growth. **d** The enantiomeric excess, ee, associated with the platelet screw dislocation handedness is driven both by the growth rate and by the intrinsic helical twist of the constituent chiral particles i.e., left-handed and right-handed for fd-wt and fdY21M, respectively. Each error bar corresponds to a confidence level of 95%, and the number of measured dislocations $N$ is indicated for each data point. The racemic conditions are marked by a red dashed line, and have been obtained either by a mixture of viral rods with opposite chiralities[24] or by an fd-wt suspension at high temperature (50 °C) for which the chiral twist vanishes[26]

dislocation, has been measured and is reported in Fig. 3d. A first analysis shows that the platelet handedness is related to both the intrinsic chirality of the constituent rod-like particles and to the kinetics of growth. Here, a direct correlation exists between the screw dislocation handedness of the platelet and the helical twist of the chiral constituent rods[33], i.e., a positive enantiomeric excess ee ≥ 0 for viral particles having a right-handed twist (fdY21M), and a majority of left-handed helical platelets (ee ≤ 0) for fd-wt rods with opposite chirality. Therefore, the prevailing handedness of the platelet screw dislocation is provided by the helical twist associated with the viral chiral constituent rods, defining the so-called chiral platelets. However, the enantiomeric excess never reaches one in the range of explored growth rates $k_r$, meaning that antichiral platelets are also formed, whose handedness is opposite to the one of the constituent viruses, as shown for instance by platelets with right-handed screw dislocation composed of left-handed fd-wt. The existence of antichiral platelets leads to a chiral

balance (ee = 0) at the lowest growth rate, regardless of the intrinsic rod chirality (Fig. 3d). Conversely, chiral platelets are more likely to be produced for the fastest growth kinetics. Control experiments of platelet formation at high growth rate have been performed in conditions where the rod chirality is vanished. For this purpose, a racemic mixture composed of viral rods with opposite chiralities has been prepared[24], as well as a suspension of fd-wt viruses at high temperature (50 °C) in which the chiral twist is suppressed[26]. In both cases, the absence of enantiomeric excess (ee = 0) confirms that without any intrinsic particle chirality, no chiral symmetry breaking occurs with the appearance of a promoted platelet handedness. Although the intrinsic rod chirality is a necessary condition for getting a prevailing handedness (ee ≠ 0), the kinetics of growth is the key parameter to continuously tune the overall sample chirality from racemic (ee = 0) to helical (|ee| > 0).

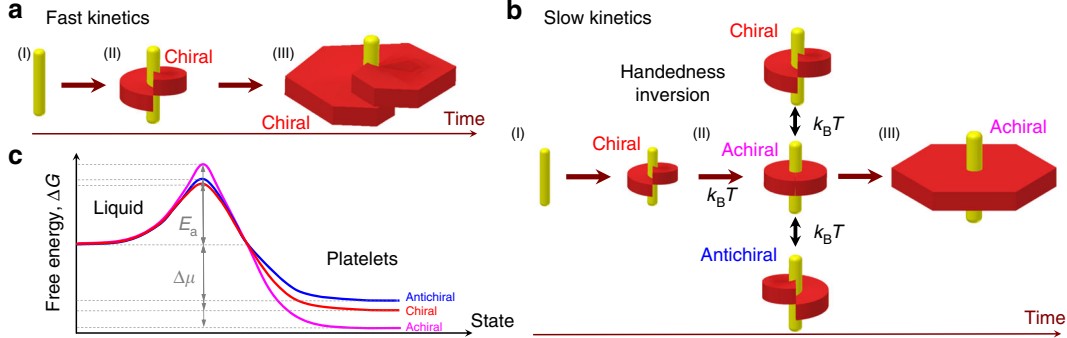

**Fig. 4** Mechanism of the platelet handedness selection process, which is driven by the interplay between kinetics of growth, thermal fluctuations, and intrinsic helical twist of the chiral constituent rods. **a** In a first step (I), the low fraction of multimeric viral rods self-assemble into needle-like nuclei (in yellow) acting as crystallization centers for the platelet formation (in red). Single viral rods condensate radially around the protruding nucleus core into helical germs (II), whose handedness is determined by the intrinsic twist of the constituent viruses. For fast kinetics, the presence of screw dislocations promotes the crystal growth, ending solely with chiral platelets (III). **b** In the opposite kinetic limit, at very slow growth rate, platelet germs undergo thermal fluctuations in the early stage of their formation (II), leading to a randomization of their helicity favoring then achiral morphologies. A few antichiral platelets, having an opposite handedness with respect to the one of the chiral forming rods, can be also produced by thermal fluctuations. In advanced growth stage, the handedness of both chiral and antichiral platelets is locked by the platelet overlap (III). **c** The crystallization is assumed to be an activated process, whose energy barrier ($E_a$) is kinetic-dependent. Despite their intrinsic frustration stemming from the lack of helical twist, the achiral platelets have the lowest free energy as compared to the chiral and a fortiori to the antichiral ones. Both chiral and antichiral platelets exhibit a screw dislocation which promotes their growth and therefore decreases their activation energy $E_a$

## Discussion

We propose a qualitative and phenomenological mechanism to account for these findings (Fig. 4a, b). The crystallization is assumed to be an activated process, whose energy barrier ($E_a$) depends on the kinetic pathway[34]. By considering an Arrhenius-type dependence for the growth rate, $k_r \sim \exp(-E_a/k_BT)$[35], the activation energy for crystallization $E_a$ decreases by increasing $k_r$ and is therefore lower for platelets exhibiting dislocations, as the presence of defects promote their fast growth (Fig. 4c). Beyond kinetics, the qualitative comparison of the free energy associated with the different platelet morphologies has to be considered. The geometric frustration of locally preferred hexagonal packing competing with the intrinsic twist of the chiral colloidal rods leads to introduction of either stress or defects in the structure[36,37]. Because of the absence of twist of the constituent particles in the 2D crystals including those lying at the edges (Fig. 1d and Supplementary Movie 1), achiral flat platelets are intrinsically frustrated. In contrast, chiral platelets exhibit twist introduced through their screw dislocation, similar to the Eshelby twist observed for nanowires[16,17]. In these structures, rods are slightly slipped over one another, allowing then the hexagonal lattice to remain locally undistorted. If in usual crystals, the (out-of-core) free energy of the dislocation increases quadratically with the magnitude of the Burgers vector ($E \sim b^2$), this energy is vanishing in the quadratic approximation ($E \sim b^4$) in lamellar structures such as the liquid crystalline smectic phases[38]. In the same vein, it is expected that the elastic deformation of a helical crystalline platelet with a topological defect only represents a small contribution to its free energy, as supported by the occurrence of antichiral platelets having opposite handedness with respect to their constituent rods. Therefore, only the energy cost associated with elastic deformation has to be considered when a screw dislocation is introduced in the colloidal platelets, as the central defect core always pre-exists whatever the 2D crystal morphology is (i.e., even for achiral flat crystallites). Despite the presence of twist decreasing their free energy, the chiral platelets, and a fortiori the antichiral ones, are expected to have a slightly higher free energy than the intrinsically frustrated achiral ones (Fig. 4c). This stems both from the (low) elastic energy associated with the

platelet deformation and from the line tension cost for creating extra edges when defects are introduced.

The platelet growth proceeds in a few distinct steps (Fig. 4a, b). After the initial nucleation and formation of the central protruding defect acting as a homogeneous seed to propagate the crystallization, radial growth takes place leading to a helical germ forming a screw dislocation, whose handedness is fixed by the intrinsic chirality of the constituent viruses (Fig. 4a). As screw dislocations make possible fast growth and provide the lowest activation energy $E_a$, this results, for fast kinetics, in chiral platelets which have conserved their primary screw dislocation.

Conversely, at very slow growth rate, platelet germs undergo thermal fluctuations, as suggested by the lack of smooth faceting during the early stage of their crystallization (Supplementary Movie 7). This behavior, reminiscent of the roughening transition in which molecules jiggle in the crystal germ[31], would allow for the inversion of the screw dislocation handedness, leading therefore to a statistical randomization of their helicity. Thermal fluctuations promote achiral structures, and make easier to cross the higher energy barrier $E_a$ (Fig. 4c). For these very slow kinetics, antichiral platelets, having an opposite handedness with respect to the constituent chiral rods, can also emerge because of the low elastic deformation energy associated with screw dislocations. At longer times, both antichiral platelets, which are not energetically favored, and the chiral ones are stabilized by the platelet overlap locking the screw dislocation handedness. Therefore interplay between growth kinetics, thermal fluctuations, and intrinsic helical twist of the chiral constituent rods drives the selection process of the topological defect handedness, from chiral to antichiral via achiral crystalline structures.

In summary, we have evidenced a screw dislocation based mechanism to monitor, through the kinetics of growth, the chiral morphology of 2D crystals from achiral to twisted structure. The control of the chirality transfer mediated by topological defects from racemic to predominantly helical is demonstrated by tuning the crystallization kinetic pathway. This provides a framework in which topological defects drive the chirality amplification from constituent particles to the resulting crystalline structures,

allowing for the design and elaboration of chiral functional materials of tunable helicity.

## Methods

**Virus suspensions.** Filamentous fd viruses were used as a model system of colloidal rod-like particles, which have shown to behave nearly as hard rods[39]. We used two different fd virus strains, fd-wt (wild-type) and fdY21M (mutant), which only differ by a single-point mutation of the 21$^{st}$ amino acid of each of the 3000 major coat proteins, changing from tyrosine in fd-wt to methionine in fdY21M[24]. This results in a cholesteric phase which exhibits opposite twist handedness for the two viral strains (left-handed for fd-wt and right-handed for fdY21M) allowing to investigate the effect of intrinsic twist of chiral particles in the platelet formation. Note that the chirality of fd-wt is temperature sensitive whereas fdY21M one is temperature independent[26]. Both viruses were grown using ER2738 strain of *E. coli* as host bacteria and purified according to standard protocols[40]. They were then dialyzed against TRIS-NaCl-HCl buffer at 110 mM of ionic strength and pH 8.2, in order to screen their electrostatic repulsion, and concentrated by using ultra-centrifugation at 30–35 mg ml$^{-1}$ in the same buffer solution. The virus concentration was determined using spectrophotometry[24]. Fluorescent virus batches were separately prepared by labeling their coat proteins either with Alexa Fluor 488-TFP ester (Invitrogen) or fluorescent Dylight550-NHS ester (ThermoFischer) dyes. For bioconjugation with the viruses, an initial excess of three fluorescent dyes per coat protein was introduced, resulting in an average value of about 1200 dye molecules per virus, as previously reported[41]. A small fraction (from 0.001% (w/w) to 0.1% (w/w) depending on the experiments) of these labeled viruses were added to the non-labeled virus batches for fluorescence microscopy observations. Because of the modification of their surface by fluorescent dyes, labeled viral particles are difficult to insert within densely packed structures, and were found to mainly decorate platelet edges (Figs. 1c, d and 2b). The virus polydispersity was estimated by transmission electron microscopy experiments using negative staining of the particles[42] from which the size distribution of their contour length was obtained (Supplementary Figure 2). The contour length L of single particles was measured to be 880 and 910 nm for fd-wt and fdY21M, respectively, leading to contour length over persistence length ratio of $L/L_P = 0.3$ and 0.1 for fd-wt and fdY21M strains[24]. Multimeric viruses (mainly dimers and a few trimers, having two and three times the nominal length of the single virus), whose probability of growth decreases by increasing their length, were present in a few percent (4% and 9% for fd-wt and fdY21M batches, respectively) in the suspension[26]. These longer rods preferentially self-assembled into long central protruding columnar bundles from which the hexagonal platelets grew.

**Sample preparation.** Non-adsorbing polymer, poly(ethylene glycol) (PEG) of average molecular weight $M_w = 8000$ g mol$^{-1}$ (Sigma-Aldrich) was added to the virus suspension to induce depletion attractive interaction between the rod-like viruses. The PEG radius of gyration $R_g$ is about 4 nm[43], and is therefore of the same magnitude as the virus rod diameter of 7 nm[40]. Aqueous mixtures of fd viruses and PEG were prepared and then immediately injected into an optical microscopy cell. This experimental cell is composed by a glass slide and a coverslip, separated by a spacer (Mylar or Parafilm) whose thickness was tuned between 8 and 120 µm. Thin cell thicknesses have shown to promote fast crystallization of the platelets. Before being sealed by ultraviolet-cured glue (NOA81, Epotecny), the glass surfaces were initially cleaned with sulfochromic acid. Some coating of the cell glass with poly-acrylamide brush chains was tested thanks to a protocol reported elsewhere[44]. The effect of the coating is mainly to limit the depletion effect induced by the cell walls, and it has only been shown to slightly affect the phase diagram (depletant concentration) when it was used. During the glue hardening under the UV lamp, the main part of the cell window was protected from the UV light by covering it with some aluminum foil to avoid any photo-bleaching of the fluorescent dyes grafted on the virus surface. For SAXS experiments, cylindrical quartz capillaries (diameter ~ 1.5 mm; Mark-Röhrchen) were filled with approximately 20 µl of virus/polymer mixtures and then sealed by flame. The capillaries were positioned vertically (gravity along the capillary main axis) for about 4 weeks in order to induce macroscopic phase separation (platelet-rich phase at the capillary bottom).

**Optical microscopy methods.** The differential interference contrast (DIC) and epifluorescence images were obtained using an inverted optical microscope (IX71, Olympus) equipped with a 100× oil-immersion objective (NA 1.4, UPLSAPO), a piezo device for objective z-positioning (P-721 PIFOC Piezo Flexure Objective Scanner, PI) operated by computer interface software (Meta-Morph, Molecular Devices), a mercury-halide excitation source (X-cite120Q, Excelitas), and fluorescence imaging cameras (CoolSnap ES, Photometrics, or NEO sCMOS, Andor Technology). To simultaneously acquire DIC and fluorescence images, an optical splitter setup (Optosplit II, Cairn Research) was used dividing each image into two channels thanks to an appropriate dichroic beamsplitter and band pass filters. The respective images were then overlaid with a precision of one pixel. The determination of the handedness of the platelet screw dislocation was performed thanks to the objective piezo z-positioner, incrementally moving (step interval: 0.2–0.3 µm) from the coverslip-side (bottom) to the slide glass-side (top) of the experimental cell. Face-on platelets were selected for these measurements, and imaged at each

step resulting in a z-stack sequence for each platelet. A platelet diameter higher than typically 5 µm was needed to enable the handedness determination by optical microscopy, limiting therefore the highest enantiomeric excess experimentally measured, as the platelet size usually decreases by increasing the growth rate $k_r$. The microscope was also equipped with a temperature controller (MK2, INSTEC) to investigate the effect of temperature (50° C) on the handedness of platelet screw dislocation. The temperature-dependent twist of fd-wt strain was used to induce high-temperature achiral suspension in which no twist is displayed. Unless explicitly mentioned, all sample preparation, time evolution, and observation occurred at room temperature.

**Small-angle X-ray scattering.** SAXS experiments were carried out at the SWING beamline at the synchrotron SOLEIL (Orsay, France) working at a wavelength of $\lambda$ = 0.0995 nm. Data were collected by an AVIEX CCD detector, placed in a vacuum detection tunnel and at a distance of 1.49 m from the sample. From the SAXS pattern, angular integration were performed to provide the scattered intensity as a function of the scattering vector modulus, $q$. The width of Bragg peaks was only limited by the instrumental resolution ($\Delta \sim 0.025$ nm$^{-1}$). In Supplementary Figure 1, the local virus concentration has been estimated according to the 2D swelling law determined for dense suspensions of fd viruses[45]: $q_{100} = \eta \times C^{1/2}$ with $\eta = $ 0.0384, $q_{100}$ the wavevector modulus of the first Bragg reflection (in nm$^{-1}$) and $C$ the rod concentration (in mg ml$^{-1}$).

**Confidence levels in screw dislocation handedness measurements.** Confidence intervals (CIs) were applied in the screw dislocation handedness measurements to account for the number of observed samples $N$, and were calculated according to the following formula, CI = $\pm \gamma$ Sp = $\pm \gamma \sqrt{\frac{p(1-p)}{N}}$, where $p$ is the sample proportion, $\gamma$ the factor determining the confidence level, and Sp the estimated standard error[46]. We used $\gamma = 1.96$ which yields a confidence level of 95%. Here $p = R/N$ and $N = L + R$ with $R$ and $L$ standing for the number of platelets exhibiting right-handed and left-handed screw dislocation, respectively. In case of enantiometric excess defined by ee = $(R - L)/(R + L)$ and reported in Fig. 3d, the standard error is twice the one defined for $p$, and this gives an error bar of $\pm 2 \times 1.96 \times$ Sp $\approx$ $\pm 4\sqrt{\frac{p(1-p)}{N}}$ for a confidence level of 95%.

**Determination of platelet growth rate $k_r$.** In order to quantitatively characterize the kinetic growth of a platelet from the protruding defect core, we used the Verhulst model based on the logistic equation given by[47]: $\frac{dD}{dt} = k_r D - \frac{k_r}{D_f} D^2 = k_r D \left(1 - \frac{D}{D_f}\right)$ (1), where $D$ is the platelet diameter at time $t$, $D_f$ the final diameter in the long-time limit, and $k_r$ the growth rate parameter. The time evolution of $D$ as initially described by an exponential growth is compensated by a feedback term scaling as $D^2$ accounting for the finite size of the sample (fixed number of rods). The analytical solution of Eq. (1) is $D(t) = \frac{D_f D_0 e^{k_r t}}{D_f + D_0 (e^{k_r t}-1)}$, where $D_0$ is the initial diameter given by the width of the needle-like defect core. The growth rate parameter, $k_r$, was then determined by numerically fitting the experimental data points (Fig. 3a).

**Data availability.** All relevant data are available from the corresponding author on request.

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

## Acknowledgements

This work was supported by the French National Agency (ANR) through the project AURORE. We thank D. Constantin for the SAXS beamtime and P. Merzeau for the drawing of the schemes.

## Author contributions

E.G. designed and supervised the study, and performed initial experimental observations. B.S. prepared the samples, and performed the experiments. A.C. performed some of the fluorescence microscopy experiments and the electron microscopy imaging. B.S. treated and analyzed the data. E.G. and B.S. wrote the manuscript.

## Additional information

**Competing interests:** The authors declare no competing interests.

