## [Peer Review File(PDF 447 kb) · Nature Communications]

Reviewers' comments:

Reviewer #1 (Remarks to the Author):

This manuscript presents an experimental study of how chirality of supramolecular structures is governed by the interplay of the constituent molecule's intrinsic chirality and growth kinetics of the macroscopic structure. Specifically, it shows how crystalline platelets of rod shaped particles develop helicity around a screw dislocation. It is a long-standing problem, particularly in liquid crystals, to determine the relationship between the microscopic chirality of the constituent rods and the macroscopic twist of the bulk phase. The authors present an interesting finding, in context of this problem, that the growth kinetics of the bulk phase is a crucial parameter in addition to the intrinsic chirality of the constituents. However, the work presented in the manuscript is incomplete and can only be accepted for publication after significant revisions. The major points of concern are as follows:

1. I find it surprising that even at the highest growth rate, the enantiomeric excess reaches at best 0.7 or so as presented in Fig. 3d. This implies that contrary to the statements in abstract and conclusion, the authors have not really observed a pure homohelical phase. At best, they have observed what can be called a predominantly homohelical phase. More than the terminology used, the nature of advance made by the authors is on a weak footing. The references cited by them (ref 7, 8) show that it is possible to achieve enantiomeric excess close to 1 for the case of molecular crystals while their own data is way short of this achievement. The authors should comment on why the colloidal system that they are dealing with is inherently limited to attaining an enantiomeric excess of 0.7 or so.

2. Figure 3c and 3d use the growth rate, k_r , as the X-axis of the plots. But how k_r has been varied in these plots has not been described clearly. k_r depends on both the initial viral rod concentration and depleting polymer concentration. So, which of these two parameter were changed to vary k_r in Fig. 3c,d. Or, is it that both parameters were changed to vary k_r in Fig. 3 c,d ?

3. The current form of the explanation proposed by the authors is qualitative and highly speculative in nature (Fig. 4b,c). Figure 4c depicts generic curves of free energy as a function of state variable for chiral, achiral and anti-chiral platelets based on the qualitative ideas presented in the main text. I have the following concerns regarding this plot:

(a) Energy barrier (E_a) being inversely proportional to supersaturation is not rigorously analogous to E_a being inversely proportional to growth rate, a kinetic variable. Combining this analogy with their experimental observations, they conclude that E_a must be lowest for chiral platelets and highest for achiral platelets. Such a strong conclusion needs to be justified at least with simple theoretical understanding/calculations.

(b) The authors simply state the way they ascribe different free energy minima to the various platelet phases in the figure caption of Fig. 4c. A few sentences justifying this attribution should be written in the main text.

(c) To summarize, the authors have proposed a mechanism in light of their experimental observations instead of a mechanism that is rooted in first principles or the unarguable fundamental microscopics of the system.

They further propose a mechanism of how platelets could switch between anti-chiral and achiral or chiral and achiral through thermal fluctuations (Fig. 4 b). This too is based loosely on qualitative observations and raises the following concern:

(d) The authors state "viral rods jiggle in the crystal germ, would allow for the inversion of the screw dislocation handedness, leading therefore to a statistical randomization of their helicity".

However, they have not provided any quantitative experimental data to show that this jiggling exists and how that leads to a switch in the helicity of the platelet. The beauty of colloid experiments is that one can often watch all the mechanisms since time scales and length scales are not prohibitive. So, they must investigate if they can experimentally establish the mechanism of switching between different kinds of platelets. In the event, that experimental observations of such kind are found unfeasible, some simple back of the envelope theoretical calculations should be presented to explain why the experimental observations of the mechanism are practically not possible.

Other detailed comments:

1. The authors should investigate whether confocal imaging of the chiral platelets can be performed to reconstruct a real 3-D image instead of being limited to z-stacks.
2. The manuscript states that there is absence of particle self-diffusion within the platelets. However, this should be supplemented with a quantification of the self-diffusion. That is, whether there is any detectable rod motion and if not, what is the resolution of the particle tracking algorithm?
3. Videos S1-S4 can easily be combined into a single video. All of them are showing DIC and fluorescence overlaid images of the same system in different configurations. In fact, the authors provide a combined caption for Videos S1-S4. The time legend on these movies is written in ms. It should be converted to seconds because though 7050 ms may look like a large number, it is merely about 7 seconds!!
4. The methods section should specify the degree of labeling of the fluorescent rods (average number of dye molecules per viral rod) used in the experiment.

Reviewer #2 (Remarks to the Author):

In the manuscript entitled "Chirality induced crystallization via screw-dislocations" the authors describe a series of experimental studies whose goal is to understand how monodisperse rodlike particles assemble into 2D crystalline membranes. The authors describe an interesting set of experimental data that demonstrate how chirality, defect formation and crystallization kinetics couple to each other to determine the complex shapes of virus based colloidal structures. The experiments are carefully done and authors have collected extensive data sets. The interpretation of data is sound. Recently, there has been a significant increase in research that is focused on understanding the self-assembly of colloidal rods in presence of attractive interactions. The submitted manuscript presents a valuable contribution to this area of research. The manuscript is well written and clear. For these reasons I recommend that the manuscript be accepted for publication.

Point-by-point response to Referee's comments

Manuscript NCOMMS-17-30582-T entitled CHIRALITY-CONTROLLED CRYSTALLIZATION VIA SCREW DISLOCATIONS by B. Sung *et al.*

We first would like to thank both Reviewers for their positive evaluation of our paper, their understanding of the significance of our work in the context of chirality transfer from constituent particle scale to supramolecular and superstructure level, and for their valuable comments which helped us to improve our manuscript. The details of the changes made in our revised manuscript have been highlighted in yellow in a separate PDF document.

Reviewer #1 (Remarks to the Author):

This manuscript presents an experimental study of how chirality of supramolecular structures is governed by the interplay of the constituent molecule's intrinsic chirality and growth kinetics of the macroscopic structure. Specifically, it shows how crystalline platelets of rod shaped particles develop helicity around a screw dislocation. It is a long-standing problem, particularly in liquid crystals, to determine the relationship between the microscopic chirality of the constituent rods and the macroscopic twist of the bulk phase. The authors present an interesting finding, in context of this problem, that the growth kinetics of the bulk phase is a crucial parameter in addition to the intrinsic chirality of the constituents. However, the work presented in the manuscript is incomplete and can only be accepted for publication after significant revisions. The major points of concern are as follows:

1. I find it surprising that even at the highest growth rate, the enantiomeric excess reaches at best 0.7 or so as presented in Fig. 3d. This implies that contrary to the statements in abstract and conclusion, the authors have not really observed a pure homohelical phase. At best, they have observed what can be called a predominantly homohelical phase. More than the terminology used, the nature of advance made by the authors is on a weak footing. The references cited by them (ref 7, 8) show that it is possible to achieve enantiomeric excess close to 1 for the case of molecular crystals while their own data is way short of this achievement. The authors should comment on why the colloidal system that they are dealing with is inherently limited to attaining an enantiomeric excess of 0.7 or so.

- We thank Reviewer #1 for bringing up this question about the limited value of our enantiomeric excess (ee). In order to avoid any ambiguity or overclaim, we have revised our manuscript according to Reviewer #1 suggestion and have added the word 'predominantly' in front of 'homochiral' in the introduction (page 1) and of 'helical' in the conclusion (page 9). To reply specifically to the point, let's first emphasize that our experiments cannot be directly compared to the ones by Kondepudi *et al.* and Viedma (refs. 7 and 8). In the latter cases, they have studied *chiral symmetry breaking* during the crystallization process, starting from a racemic state (as sodium chlorate formed by achiral molecules). This results after crystallization in a *random* enantiomeric excess of either +1 or -1, with an equal probability. In the work presented here, this is somehow an opposite case: we start from a *chiral* system of particles (viruses being either left- or right-handed and fixing accordingly the sign of ee) for which we control the *degree of racemization* of the resulting crystals by tuning the kinetics of growth. Therefore, we do not see any chiral symmetry breaking with enantiomeric excesses having possibly opposite signs: for a given set of *chiral* particles, the enantiomeric excess of the associated crystals can be continuously tuned, from 0 (racemic) to a finite value ($ee < 1$) whose sign is predetermined by the chirality of the constituent particles. The fact that we didn't attain the highest value $|ee| = 1$ mainly relies on an experimental limitation: below a platelet diameter of about 5 μm , it was practically found very difficult to

determine by optical microscopy the screw dislocation handedness. An example of such a small platelet of about 3 μm in diameter is provided below, where a Z-stack of images illustrates the difficulty associated with the handedness determination (scale bar: 1 μm):

Moreover, the general trend is that the faster the growth is, the smaller the crystallites are. Despite many tries, we didn't succeed to find a set of experimental conditions (both in virus and polymer concentrations, see point 2 below for more details) reaching higher ee than those reported in Figure 3d, i.e. both very fast growth rate k_r AND platelet size big enough to enable the dislocation handedness determination by optical microscopy (Note that, in case of by Kondepudi *et al.* and Viedma experiments, the chirality of sodium chlorate crystals is easily determined only by observing their change of color between slightly uncrossed polarizers, method which cannot be applied here).

Therefore, we do think that there is *no intrinsic limitation* to reach $|ee|=1$, but only extrinsic constrains related to our experimental conditions and to the optical method used to determine the handedness at the single platelet level. For making this point clearer, two changes have been added page 6 and the following sentence has been added in the 'Methods' section, page 23 of the paper: "A platelet diameter higher than typically 5 μm was needed to enable the handedness determination by optical microscopy, limiting therefore the highest enantiomeric excess experimentally measured, as the platelet size usually decreases by increasing the growth rate k_r ."

2. Figure 3c and 3d use the growth rate, k_r , as the X-axis of the plots. But how k_r has been varied in these plots has not been described clearly. k_r depends on both the initial viral rod concentration and depleting polymer concentration. So, which of these two parameter were changed to vary k_r in Fig. 3c,d. Or, is it that both parameters were changed to vary k_r in Fig. 3 c,d ?

- As noticed by Reviewer #1, *both* the initial viral rod concentration *and* the depleting polymer concentration have been varied in order to get the highest range of growth rates for which the screw dislocation handedness determination can still be performed unambiguously by optical microscopy (implying, from a practical point of view, a platelet diameter above 5 μm). This is illustrated in Figure 3a, where the black, red, blue and pink curves show an increase of k_r (in the low range of values) by increasing the virus concentration from 1 to 7 mg/mL at fixed depletant concentration ($C_{\text{PEG}}=25\text{mg/mL}$), while the navy curve on the left shows that the increase of polymer concentration to 34mg/mL (at virus concentration of 8mg/mL) strongly increase the kinetics of growth.

Both values of virus and polymer concentration are limited experimentally as shown in the phase diagram in Figure 1g: increasing too much the viral rod concentration leads to the nematic phase, while at too high depletant value, crystalline platelets are not stable anymore and slender morphologies (fibers and bundles) appear.

These are the reasons why we chose k_r as main physical parameter in Figures 3c and d to quantitatively account for the dynamics of the crystallization process, regardless of the details of the experimental conditions which have been used (polymer and virus concentrations, cell surface treatment, etc...).

3. *The current form of the explanation proposed by the authors is qualitative and highly speculative in nature (Fig. 4b,c).*

- We agree with Reviewer #1 that a whole quantitative theoretical model has still to be proposed and achieved to account for the detailed description of our experimental results.

Figure 4c depicts generic curves of free energy as a function of state variable for chiral, achiral and anti-chiral platelets based on the qualitative ideas presented in the main text. I have the following concerns regarding this plot:

(a) Energy barrier (E_a) being inversely proportional to supersaturation is not rigorously analogous to E_a being inversely proportional to growth rate, a kinetic variable. Combining this analogy with their experimental observations, they conclude that E_a must be lowest for chiral platelets and highest for achiral platelets. Such a strong conclusion needs to be justified at least with simple theoretical understanding/calculations.

- As pointed out by Reviewer #1, the analogy with classical nucleation theory is, strictly speaking, not correct. So we have rewritten this point: “The crystallization is assumed to be an activated process, whose energy barrier (E_a) depends on the kinetic pathway. By considering an Arrhenius-type dependence for the growth rate, $k_r \sim \exp(-E_a/k_B T)$, the activation energy for crystallization E_a decreases by increasing k_r and is therefore lower for platelets exhibiting dislocations, as the presence of defects promote their fast growth (Fig. 4c).”
The manuscript has been modified accordingly page 7.

(b) The authors simply state the way they ascribe different free energy minima to the various platelet phases in the figure caption of Fig. 4c. A few sentences justifying this attribution should be written in the main text.

- As suggested by Reviewer #1, we have added in the main text a few extra sentences to complete discussion of the free energy associated with the different platelet morphologies. See page 8 of the manuscript: “Despite the presence of twist decreasing their free energy, the chiral platelets, and *a fortiori* the antichiral ones, are expected to have a slightly higher free energy than the intrinsically frustrated achiral ones (Fig. 4c). This stems both from the (low) elastic energy associated with the platelet deformation and from the line tension cost for creating extra edges when defects are introduced.”

(c) To summarize, the authors have proposed a mechanism in light of their experimental observations instead of a mechanism that is rooted in first principles or the unarguable fundamental microscopics of the system.

We are not sure to understand what Reviewer #1 means by his/her remark. We have indeed proposed a mechanism based on our experimental observation, which is qualitative and phenomenological (We have added these two terms page 7 of the manuscript). We therefore expect that the possible publication of our article in Nature Communications could offer a high visibility of our work, which could stimulate further theoretical investigation of our experimental results.

They further propose a mechanism of how platelets could switch between anti-chiral and achiral or chiral and achiral through thermal fluctuations (Fig. 4 b). This too is based loosely on qualitative observations and raises the following concern:

(d) The authors state “viral rods jiggle in the crystal germ, would allow for the inversion of the screw dislocation handedness, leading therefore to a statistical randomization of their helicity”. However, they have not provided any quantitative experimental data to show that this jiggling exists and how

that leads to a switch in the helicity of the platelet. The beauty of colloid experiments is that one can often watch all the mechanisms since time scales and length scales are not prohibitive. So, they must investigate if they can experimentally establish the mechanism of switching between different kinds of platelets. In the event, that experimental observations of such kind are found unfeasible, some simple back of the envelope theoretical calculations should be presented to explain why the experimental observations of the mechanism are practically not possible.

- Our exact statement was the following: “[...] at very slow growth rate, platelet germs undergo thermal fluctuations, as *suggested* by the lack of smooth faceting during the early stage of their crystallization (Video S7). This behavior, *reminiscent* of the roughening transition in which *viral rods* (-> replaced by ‘molecules’ in the new version) jiggle in the crystal germ, *would* allow for the inversion of the screw dislocation handedness [...]”. If we agree with Reviewer #1 that our wording was certainly awkward and could be confusing, we did not intend to claim about any experimental demonstration of viral rod jiggling, as shown by the use of the word ‘reminiscent’ and the conditional tense. Our purpose was only to mention a possible analogy with the roughening transition based on the absence of clear faceting in the early stage of growth. To avoid any ambiguity, we have replaced ‘viral rods’ by ‘molecules’ in the sentence above (page 8 of the paper) to clearly state that we only mention a possible analogy. Experimentally, we have already tried to evidence any switching of handedness between the different kinds of platelets, focusing on the early stage of the crystal germ growth. However, we unfortunately failed in our investigations, because of the intrinsic limited resolution of optical microscopy (0.25 to 0.5 μm depending on the direction) which does not allow for a precise tracking of small germs in 4D (the smaller the germs are, the more they diffuse according to their Stokes diffusion coefficient). Moreover, the use of fluorescently labeled viral particles to probe the behavior inside the crystal germ at the single particle scale did not work also here because, as mentioned in the ‘Methods’ section pages 20 & 21, the labeled particles are mainly expelled from these dense crystals (as their surface properties are different), only decorating platelet edges at the end of the growth process. Therefore, no further information can be obtained by this optical approach at the single particle level. In summary, we currently do not have any available experimental tools to study the time evolution of the crystals germs in the early stage of their growth, but this point would certainly deserve further investigations.

Other detailed comments:

1. The authors should investigate whether confocal imaging of the chiral platelets can be performed to reconstruct a real 3-D image instead of being limited to z-stacks.

- As suggested by Reviewer #1, we worked on 3D reconstruction of our chiral platelets and an example is provided below. Note that even when using confocal microscopy, the 3D reconstruction is always performed thanks to a Z-stack of images, which compose the raw data. Because we are not convinced that such a representation really helps to distinguish the handedness of our platelet, we decided not to include it in the current version of the manuscript.

2. *The manuscript states that there is absence of particle self-diffusion within the platelets. However, this should be supplemented with a quantification of the self-diffusion. That is, whether there is any detectable rod motion and if not, what is the resolution of the particle tracking algorithm?*

- We would like to emphasize that we prove the crystalline structure of the platelets by SAXS (Fig. S1), and that the absence of *any detectable* (these two words have been added in the manuscript, page 3 and in the caption of Videos S1-S3 page 27) particle self-diffusion is just consistent with this statement.

To reply to Reviewer #1 concern, the resolution of our particle tracking algorithm is typically ± 1 pixel which corresponds to a length scale of 0.13 μm . Note that we have a resolution slightly higher than the standard optical resolution of the microscope, because of the numerical fit method (involving a few pixels) used for determining the particle position.

3. *Videos S1-S4 can easily be combined into a single video. All of them are showing DIC and fluorescence overlaid images of the same system in different configurations. In fact, the authors provide a combined caption for Videos S1-S4. The time legend on these movies is written in ms. It should be converted to seconds because though 7050 ms may look like a large number, it is merely about 7 seconds!!*

- We have followed Reviewer #1 suggestion and limited our number of movies to two (instead of four), keeping only edge-on views of the platelets: one focusing on the platelet itself, and one on the central protruding defect core. The time unit is expressed in ms, because beyond the overall duration of the movie which is somehow arbitrary, it is indicative of the frame rate used for recording the movies (typically between 15 and 33 fps) and it provides then the typical time scale associated with our single particle tracking experiments.

4. *The methods section should specify the degree of labeling of the fluorescent rods (average number of dye molecules per viral rod) used in the experiment.*

- Page 20 of the paper, in the Methods section, the following sentence has been added, as well as the reference 41: “For bioconjugation with the viruses, an initial excess of 3 fluorescent dyes per coat protein was introduced, resulting in an average value of about 1200 dye molecules per virus, as previously reported [41].”

Reviewer #2 (Remarks to the Author):

In the manuscript entitled “Chirality induced crystallization via screw-dislocations” the authors describe a series of experimental studies whose goal is to understand how monodisperse rodlike particles assemble into 2D crystalline membranes. The authors describe an interesting set of experimental data that demonstrate how chirality, defect formation and crystallization kinetics couple to each other to determine the complex shapes of virus based colloidal structures. The experiments are carefully done and authors have collected extensive data sets. The interpretation of data is sound. Recently, there has been a significant increase in research that is focused on understanding the self-assembly of colloidal rods in presence of attractive interactions. The submitted manuscript presents a valuable contribution to this area of research. The manuscript is well written and clear. For these reasons I recommend that the manuscript be accepted for publication.

- We thank Reviewer #2 very much for his/her strong recommendation, and for supporting the publication of our paper in *Nature Communications*.

REVIEWERS' COMMENTS:

Reviewer #1 (Remarks to the Author):

The authors have addressed my concerns satisfactorily in the revised version of the manuscript. I recommend the revised version for publication.